# Time to revisit the endpoint dilution assay and to replace the TCID$_{50}$ as a measure of a virus sample's infection concentration

**Daniel Cresta**[1☯¤], **Donald C. Warren**[2☯], **Christian Quirouette**[1], **Amanda P. Smith**[3], **Lindey C. Lane**[3], **Amber M. Smith**[3], **Catherine A. A. Beauchemin**[1,2]*

**1** Department of Physics, Ryerson University, Toronto, Ontario, Canada, **2** Interdisciplinary Theoretical and Mathematical Sciences (iTHEMS) program, RIKEN, Wako-shi, Saitama, Japan, **3** Department of Pediatrics, University of Tennessee Health Science Center, Memphis, Tennessee, United States of America

☯ These authors contributed equally to this work.
¤ Current address: Department of Physics and Astronomy, Western University, London, Ontario, Canada
* cbeau@ryerson.ca

**Data Availability Statement:** The authors confirm that all data underlying the findings are fully available without restriction. The code is freely

## Abstract

The endpoint dilution assay's output, the 50% infectious dose (ID$_{50}$), is calculated using the Reed-Muench or Spearman-Kärber mathematical approximations, which are biased and often miscalculated. We introduce a replacement for the ID$_{50}$ that we call Specific INfection (SIN) along with a free and open-source web-application, **midSIN** (https://midsin.physics.ryerson.ca) to calculate it. **midSIN** computes a virus sample's SIN concentration using Bayesian inference based on the results of a standard endpoint dilution assay, and requires no changes to current experimental protocols. We analyzed influenza and respiratory syncytial virus samples using **midSIN** and demonstrated that the SIN/mL reliably corresponds to the number of infections a sample will cause per mL. It can therefore be used directly to achieve a desired multiplicity of infection, similarly to how plaque or focus forming units (PFU, FFU) are used. **midSIN**'s estimates are shown to be more accurate and robust than the Reed-Muench and Spearman-Kärber approximations. The impact of endpoint dilution plate design choices (dilution factor, replicates per dilution) on measurement accuracy is also explored. The simplicity of SIN as a measure and the greater accuracy provided by **midSIN** make them an easy and superior replacement for the TCID$_{50}$ and other *in vitro* culture ID$_{50}$ measures. We hope to see their universal adoption to measure the infectivity of virus samples.

## Author summary

The infectivity of a virus sample is measured by the infections it causes. One approach, the endpoint dilution assay, aims to estimate the number of TCID$_{50}$ contained in a sample, where one TCID$_{50}$ is the dose at which a virus sample is expected to infect a tissue or cell culture 50% of the time, on average. Unfortunately, the commonly used methods to estimate the TCID$_{50}$ from the assay's outcome yield biased approximations that relate poorly to the number of infections the sample will cause. We propose replacing the TCID$_{50}$ with

available on GitHub (https://github.com/cbeauc/midSIN) and the midSIN tool is available as a web application (https://midsin.physics.ryerson.ca).

**Funding:** This work was supported in part by Discovery Grant 355837-2013 (CAAB) from the Natural Sciences and Engineering Research Council of Canada (www.nserc-crsng.gc.ca), Early Researcher Award ER13-09-040 (CAAB) from the Ministry of Research and Innovation of the Government of Ontario (www.ontario.ca/page/early-researcher-awards), by the Interdisciplinary Theoretical and Mathematical Sciences programme (iTHEMS, ithems.riken.jp) at RIKEN (CAAB), and by R01 AI139088 (AMS, APS, LCL) from the NIH NIAID (www.niaid.nih.gov). The funders had no role in study design, data collection and analysis, decision to publish, or preparation of the manuscript.

**Competing interests:** The authors have declared that no competing interests exist.

a more accurate, robust, and biologically meaningful measurement unit we call Specific INfection (SIN). It corresponds to the number of infections the virus sample will cause, which can be used directly to achieve the desired multiplicity of infection. Computing the SIN from one's endpoint dilution assay outcome requires no change in experimental procedure, and can be done conveniently via a web-application we developed, called **midSIN**. **midSIN** can be accessed for free on any device (laptop, cellular phone, tablet) from any web browser, without the need to download and install software.

## Introduction

The progression of a virus infection *in vivo* or *in vitro*, or the effectiveness of therapeutic interventions in reducing viral loads, are monitored over time through sample collections to measure changes (increases or decreases) in virus concentrations. As such, accurate measurement of the concentration in a sample is critical to study and manage virus infections.

Methods to count infectious virus are based on counting the infections they cause, rather than the particles themselves. In practice, however, not all infection-competent virions contained in a sample will go on to successfully cause infection. Experimental conditions, cell type used or temperature or acidity of the medium, can alter the rate at which virions, that were infection-competent in the sample, will lose infectivity before they can cause infection and thus be counted. This is why, hereafter, we will refer to the quantity measured by infectivity assays as the *infection concentration* or the number of infections the sample will cause per unit volume, rather than its concentration of infectious virions, which is not a measurable quantity. Two main types of assays are used to quantify the infection concentration within a virus sample: (1) the plaque forming (PFU) or focus forming (FFU) assays; and (2) assays we will collectively refer to as endpoint dilution (ED) assays, which include the 50% tissue culture infectious dose (TCID$_{50}$), or cell culture infectious dose (CCID$_{50}$) or egg infectious dose (EID$_{50}$) assays, etc. Herein, we focus on ED assays. Technically, the plaque and focus forming assays are also endpoint dilution assays because they rely on the counting of plaques or foci (the endpoint) as a function of dilutions. However, herein, we will refer to them as plaque or foci forming assays rather than endpoint dilution assays.

The ED assay has one major, remediable weakness: its output quantity, the TCID$_{50}$ (or CCID$_{50}$ or EID$_{50}$), does not directly correspond, or trivially relate, to causing one infected cell. The simplistic calculations, introduced by Spearman-Kärber (SK) [1, 2] and Reed-Muench (RM) [3] nearly a century ago, remain the most commonly used methods to quantify a virus sample's infectivity in units of TCID$_{50}$ (or CCID$_{50}$ or EID$_{50}$) using the ED assay. Many research groups rely on spreadsheet calculators that are passed down through generations of trainees or found on the internet, and can contain errors (e.g., versions 2 and 3 of the spreadsheet calculator provided by the Lindenbach Lab at Yale University (http://lindenbachlab.org/resources.html), which have since been removed). Theoretically, a dose of 1 TCID$_{50}$ is expected to cause $-1/\ln(50\%) = 1.44$ infections [4]. However, the approximation used by the RM and SK methods introduces an often overlooked bias where 1 TCID$_{50} \approx 1.781$ infections where $1.781 = e^{\gamma}$ and $\gamma = 0.5772$ is the Euler-Mascheroni constant [5, 6]. This makes it problematic to experimentally achieve the desired multiplicity of infection when inoculating from a sample quantified via the RM or SK methods. Many have proposed replacements for the RM and SK calculations based on logit or probit transforms of the data [4, 6, 7] or on statistical analysis of the ED assay output [7, 8] with some implemented as website applications [9, 10]. Sadly, none of these improvements were widely adopted to improve estimates of the TCID$_{50}$,

possibly due to a lack of visibility of these publications, or the lack of widespread awareness of the limitations of the RM and SK methods. None proposed replacing the TCID$_{50}$ measurement unit, with a more meaningful measure.

The work herein proposes to:

1. Encourage the use of the ED assay (e.g., TCID$_{50}$ assay), but replace its output, the TCID$_{50}$/mL (or CCID$_{50}$/mL, EID$_{50}$/mL, etc.), with a new quantity in units of **S**pecific **IN**fections or SIN/mL which corresponds to the number of infections the sample will cause per mL. The word *specific* highlights the fact that the infectivity of a sample is specific to the particulars of the experimental conditions (temperature, medium, cell type, incubation time, etc.).

2. Replace the Reed-Muench and Spearman-Kärber approximations with a computer software, **midSIN** (**m**easure of **i**nfectious **d**ose in **SIN**), that relies on Bayesian inference to measure the SIN/mL of a virus sample. To avoid calculation errors and make the new method widely accessible, **midSIN** is maintained and distributed as free, open-source software on GitHub (https://github.com/cbeauc/midSIN) for user installation, but also via a free-to-use website application (https://midsin.physics.ryerson.ca) with an intuitive user interface.

Here, we present examples of **midSIN** being used to analyze influenza and respiratory syncytial virus samples. We demonstrate that **midSIN**'s output, SIN/mL, is an accurate estimate of the number of infections the sample will cause per unit volume. We show how the accuracy of the SIN concentration estimate can be controlled by experimental choice of plate layout, including the dilution factor, and the number of replicates per dilution. We compare **midSIN**'s performance to that of the RM and SK methods, and demonstrate how the latter estimators are inaccurate under various circumstances, underlining the need to adopt **midSIN** to quantify virus samples via the ED assay.

## Results

### Key features of midSIN's output

Let us consider a fictitious ED experiment, with 11 dilutions and 8 replicate wells per dilutions, in which the minimum sample dilution, $\mathcal{D}_1 = 1/100 = 10^{-2}$, is serially diluted by a factor of $10^{-0.5} \approx 0.32$ ($\mathcal{D}_2 = 10^{-2.5}$, $\mathcal{D}_3 = 10^{-3}$, ..., $\mathcal{D}_{11} = 10^{-7}$), and the total volume of inoculum (diluted virus sample + dilutant) placed in each well is $V_{\text{inoc}} = 0.1$ mL. Now, consider that a virus sample is measured using this ED experiment and one observes (8,8,8,8,8,7,7,5,2,0,0) infected wells out of 8 replicates at each of the 11 dilutions, as illustrated in Fig 1A.

**midSIN** provides a graphical output of its results, shown in Fig 1B and 1C for this example. Note how the posterior distribution for $\log_{10}(\text{SIN/mL})$ (Fig 1B) is approximately a normal distribution. This is why $\log_{10}$ of the infection concentration should be used and reported, rather than the concentration itself. **midSIN** also graphically compares the number of infected wells observed experimentally (Fig 1C, black dots) against the theoretically expected values (blue curve and grey CI bands). This graphical representation makes it easy to identify issues with the data entered or with the experiment itself.

Importantly, **midSIN** provides a more useful quantity to the user than the TCID$_{50}$: an estimate of the concentration of infections the sample will cause, SIN/mL. For this example, the concentration is $10^{6.2\pm0.1}$ SIN/mL, where 6.2 is the mode (most likely value) of $\log_{10}(\text{SIN/mL})$, and ±0.1 is its 68% credible interval (CI). The SIN/mL corresponds to the number of infections that will be caused per mL of the sample, which can be directly used to determine the sample dilution required to obtain a desired multiplicity of infection (MOI).

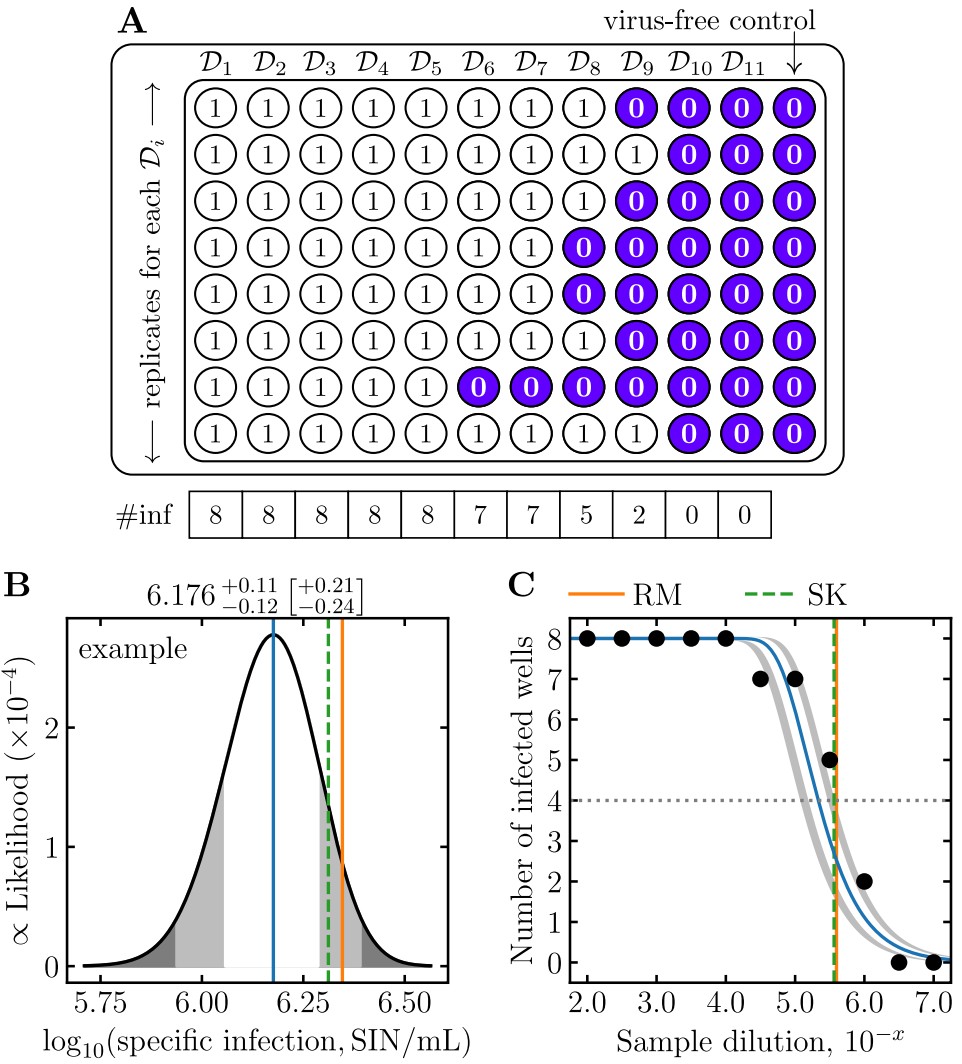

**Fig 1. Visual representation of midSIN's output for the example ED plate.** A: Illustration of the example ED plate where $\mathcal{D}_i$ are the chosen serial dilutions of the sample. For the example described in the text, $\mathcal{D}_1 = 10^{-2}$, $\mathcal{D}_2 = 10^{-2.5}$, ..., $\mathcal{D}_{11} = 10^{-7}$, with 8 replicates per dilution. The number of infected wells (# inf) is indicated at the bottom of each dilution column. B: The **midSIN**-estimated posterior distribution of the log$_{10}$ infection concentration, log$_{10}$(SIN/mL), for the example ED experiment. The vertical lines correspond to log$_{10}$(SIN/mL), based on the most likely value (mode) of **midSIN**'s posterior distribution (solid blue), or computed from the RM (solid orange) and SK (dashed green) approximations of the log$_{10}$(TCID$_{50}$) (see Methods). The $x$-value of the white and light grey region on either sides of the mode indicate the edges of the 68% and 95% credible interval (CI), respectively. The **midSIN**-estimated log$_{10}$(SIN/mL) mode ± 68% [±95%] CI are indicated numerically above the graph. C: The number of infected wells (black circles) out of the 8 replicates, as a function of the 11 serial dilutions of the example ED plate, from the least (leftmost) to the most (rightmost) diluted. For example, $x = 3.0$ corresponds to a sample dilution of $10^{-3}$ or 1/1,000. The average (expected) number of infected wells, as a function of sample dilution, is shown for the most likely value of log$_{10}$(SIN/mL) (blue curve) or its 68% and 95% CI (inner and outer edge of the grey bands, respectively). The sample dilution ($x$-value) at which the blue curve crosses the horizontal dotted line (50% infected wells) corresponds to a concentration of 1 TCID$_{50}$ per ED well volume. The vertical lines indicate the sample dilution that yields a concentration of 1 TCID$_{50}$ according to the RM and SK approximations.

In a laboratory setting, ED experiments can be performed in batches, such as to quantify the infectious concentration in samples collected at several time points over the course of a cell culture infection. For such applications, **midSIN** provides a comma separated value (csv) template file readily editable in a spreadsheet program, to collect and submit the results for batch

processing. Details on the format of the template file are available on **midSIN**'s website (https://midsin.physics.ryerson.ca). Fig 2 illustrates the output for a subset of measurements for *in vitro* infection with the respiratory syncytial virus (RSV). Each sample was measured twice, and **midSIN**'s estimates are in good agreement with one another (within 95% CI).

The *y*-axis in the left graph panels of **midSIN**'s graphical output is the non-normalized scale of the posterior distribution for $\log_{10}(\text{SIN/mL})$, which ranges between $10^{-7}$ and $10^{-2}$. The

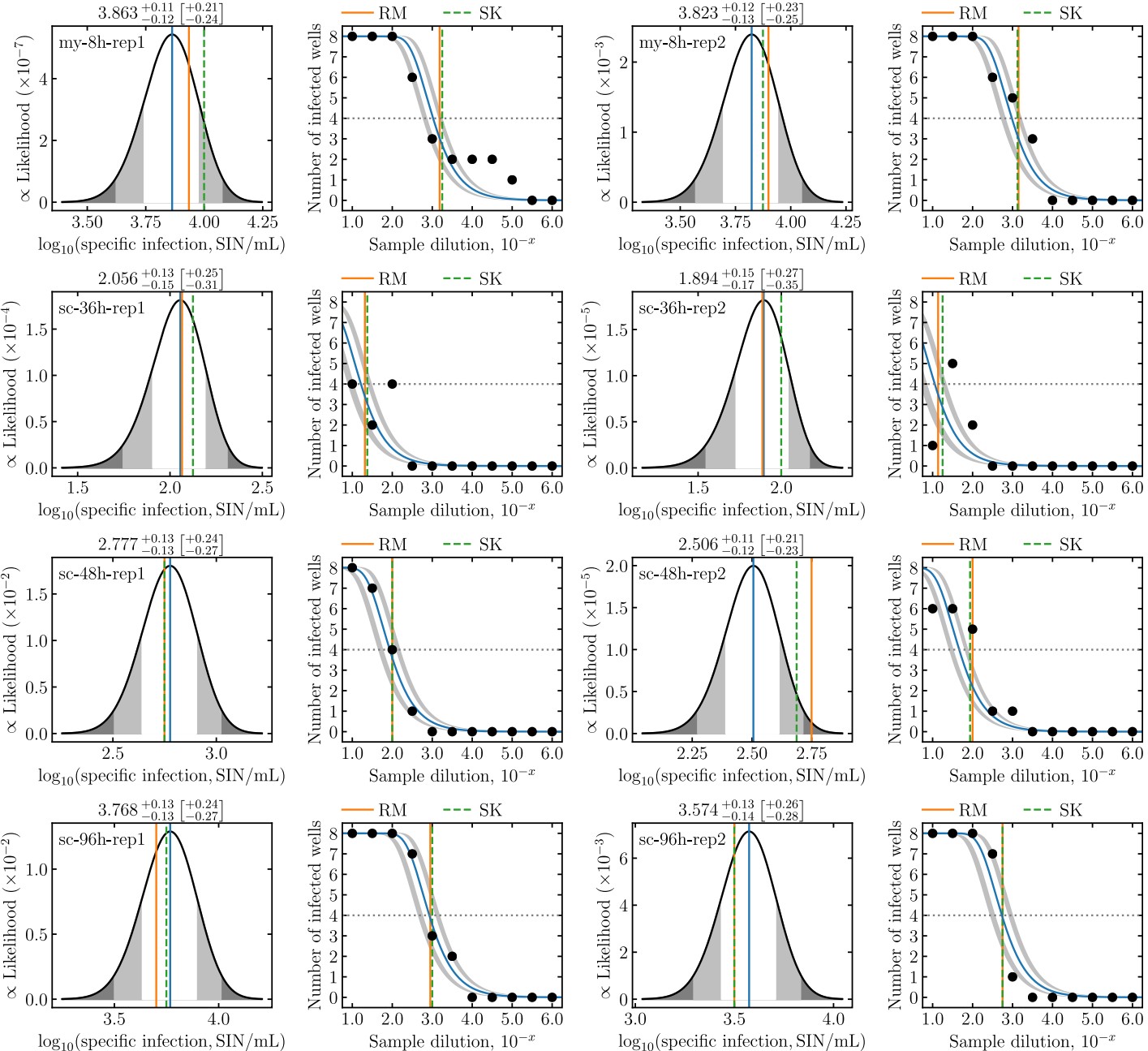

**Fig 2. Quantification of RSV sampled from *in vitro* infections.** Each row corresponds to a different experiment (mock-yield [my] or single-cycle [sc]) and sampling time point (e.g., 8 h, 36 h), and each sample was measured in duplicate (rep1, rep2). These data were collected from *in vitro* infections with the RSV A Long strain, and were previously reported in [11]. The ED measurement experiment were conducted using a plate layout of 11 dilutions, with 8 replicates per dilution, an inoculum volume of $V_{\text{inoc}} = 0.1$ mL, serial dilutions from $\mathcal{D}_1 = 10^{-1}$ to $\mathcal{D}_{11} = 10^{-6}$, separated by a dilution factor of $10^{-0.5}$.

scale loosely relates to the likelihood of observing a particular ED experimental outcome (see Methods). Unlikely ED outcomes appear as large departures of the observed number of infected wells (right panels, black dots) from what is theoretically expected (right panels, curve). It is interesting that the uncertainty (CI) of **midSIN**'s estimated $\log_{10}$(SIN/mL) appears to be independent of how much the ED outcome deviates from theoretical expectations. That is, the accuracy of **midSIN** is not strongly affected even when it is provided more unlikely, noisy experimental data. This robustness is explored further below.

## Comparing SIN to TCID$_{50}$ and PFU virus sample concentrations

The **midSIN** calculator provides an estimate of the number of infections that will be caused per mL of a virus sample (SIN/mL). In principle, a plaque assay also measures the number of infections a sample will cause, with each infection expected to develop into a plaque. If a plaque assay is performed under experimental conditions and protocols as similar as possible to those of the ED assay (i.e., using the same cells, medium, period of incubation, rinsing method, etc.), **midSIN**'s SIN/mL estimate is expected to be comparable, in theory, to the number of PFU/mL observed in the plaque assay. In practice, however, the plaque assay likely provides a biased estimate of the true concentration of infections in a sample due to various experimental limitations (e.g., distinguishing between two merged plaque and a larger one, or between small plaques and staining artifacts). To evaluate **midSIN**'s performance compared to existing methods, the infection concentration in two influenza A (H1N1) virus strain samples were measured via both plaque and ED assays, and their concentration in units of PFU, TCID$_{50}$, and SIN were compared (Fig 3). Details regarding the samples, and how the plaque and ED assays were performed are provided in Methods.

The TCID$_{50}$ concentrations estimated via the RM and SK methods are $\sim$1.5–1.7 times larger (Fig 3C and 3D) than the SIN concentration, and the set of ratios are statistically inconsistent with the assumption of equality (*p*-value: 0.01–0.03). Theoretically, 1 TCID$_{50}$ is expected to cause 1.44 infections (= 1/ln(2)) [4]. However, the RM or SK approximations are known to introduce a bias such that 1 TCID$_{50}$ estimated by these methods is expected to cause 1.781 infections (= $e^{\gamma}$ where $\gamma$ = 0.5772 is the Euler-Mascheroni constant) [5, 6]. Using the RM, SK, and SIN measurements presented in Fig 3A and 3B, we confirmed (the mean $\log_{10}$(ratio) was re-computed for ratio = (RM/1.781)/SIN and (SK/1.781)/SIN, and found to be 0.85–0.93, which is statistically consistent (*p*-value: 0.1–0.3) with the assumption of equality, i.e., ratio = $10^{0}$ = 1.) that 1.781 SIN $\approx$ 1 TCID$_{50}$ when the latter is estimated via the RM or SK approximations, as expected theoretically if SIN is indeed measuring the infection concentration in a sample.

Similarly, the ratio of the PFU concentration determined via the plaque assay and the SIN concentrations estimated by **midSIN** is $\sim$0.89–0.93, which is statistically consistent with the assumption of equality (*p*-value: 0.2–0.5). These results confirm the theoretical expectation that 1 PFU $\approx$ 1 SIN when the plaque and ED assays are performed in the same manner, as was the case here. This provides further support, via two independent assays, that the SIN concentration estimated by **midSIN** from the ED assay is a robust measure of the infection concentration of a virus sample.

## Comparing midSIN's performance to that of the RM and SK methods

The RM and SK methods rely on the number of infected wells decreasing as dilution increases. Their estimates are affected when the number of infected wells remains unchanged or even increases as dilution increases, which statistics and experimental data herein (Fig 2) tell us can reasonably occur experimentally. The RM and SK methods also mostly require that at the

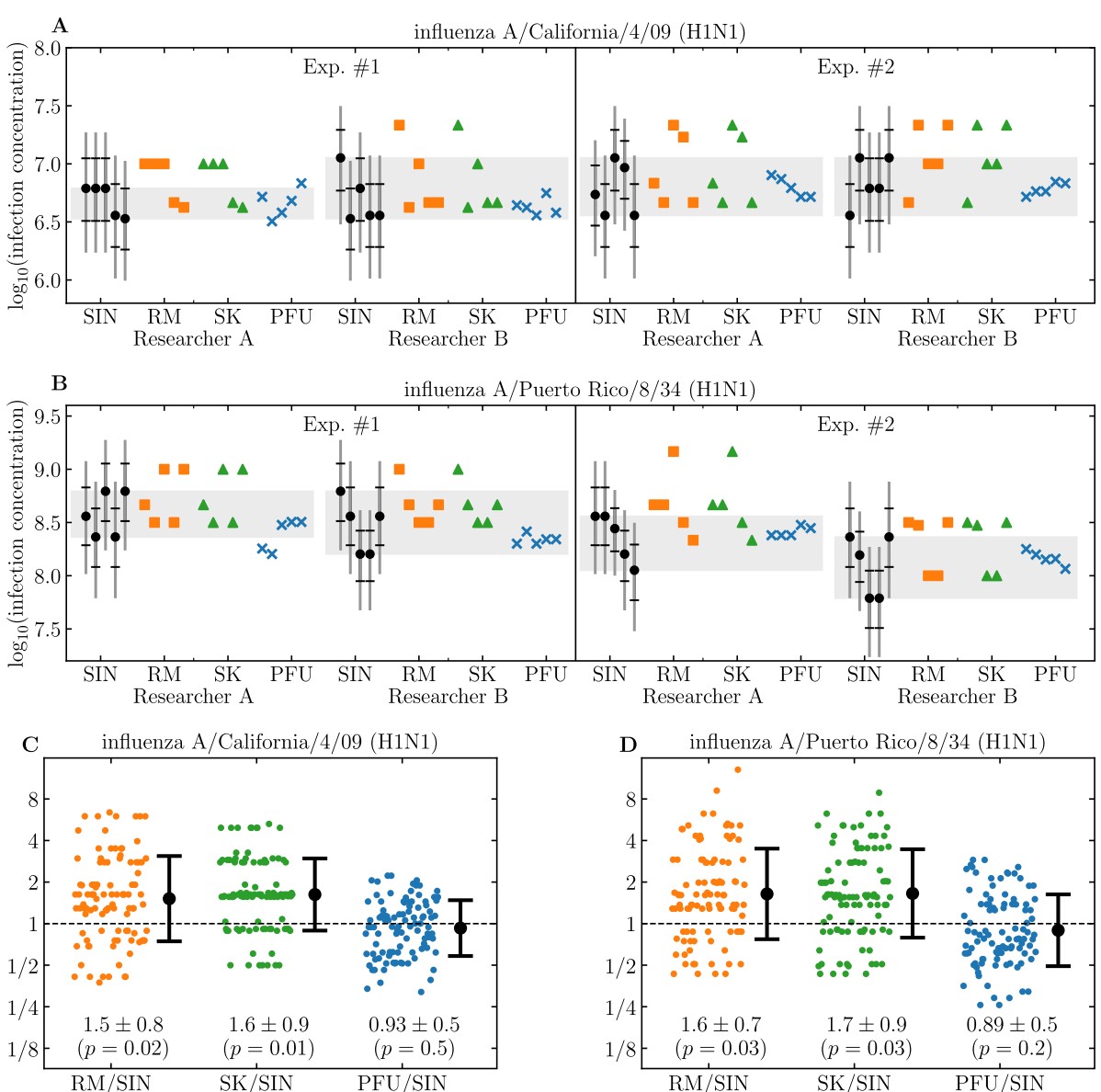

**Fig 3. Comparing SIN to TCID$_{50}$ and PFU for influenza A virus samples.** A,B: The infection concentration in two influenza A (H1N1) virus strain samples was measured via both an ED assay and a plaque assay (x, PFU). The ED assay was quantified in log$_{10}$(TCID$_{50}$) using the RM (square) or SK (triangle) methods, or in log$_{10}$(SIN) using **midSIN** (circle with 68%,95% CI). Each of the 2 strain samples was measured over 2 separate experiments (Exp. #1, #2), performed each time by 2 different researchers (Researcher A or B), with 5 biological replicates each. The grey bars indicate the range of log$_{10}$(SIN) values across the 5 replicates. The RM, SK, and SIN measures were estimated for each replicate based on the same ED plate. The experimental details are provided in Methods. C,D: The log$_{10}$ of the ratio between either the TCID$_{50}$ via the RM or SK method or the PFU, over the SIN via **midSIN**. The ratios were computed for each replicate (5 × 5 replicates), per experiment, per researcher (25 replicates × 2 researchers × 2 experiments = 100 ratios) shown as individual symbols (dots) for each method (RM, SK, PFU). The mean and 68% CI of the 100 ratios are indicated numerically and as black circles with error bars. The $p$-value indicates whether the ratios are statistically different from unity.

lowest and highest sample dilutions, all wells be infected and uninfected, respectively. Fig 4 provides a graphical representation of how the RM and SK methods estimate the TCID$_{50}$ concentration from an ED assay. Simply stated, the RM and SK methods use geometric arguments to estimate the sample dilution at which 50% of wells would be infected. While they are

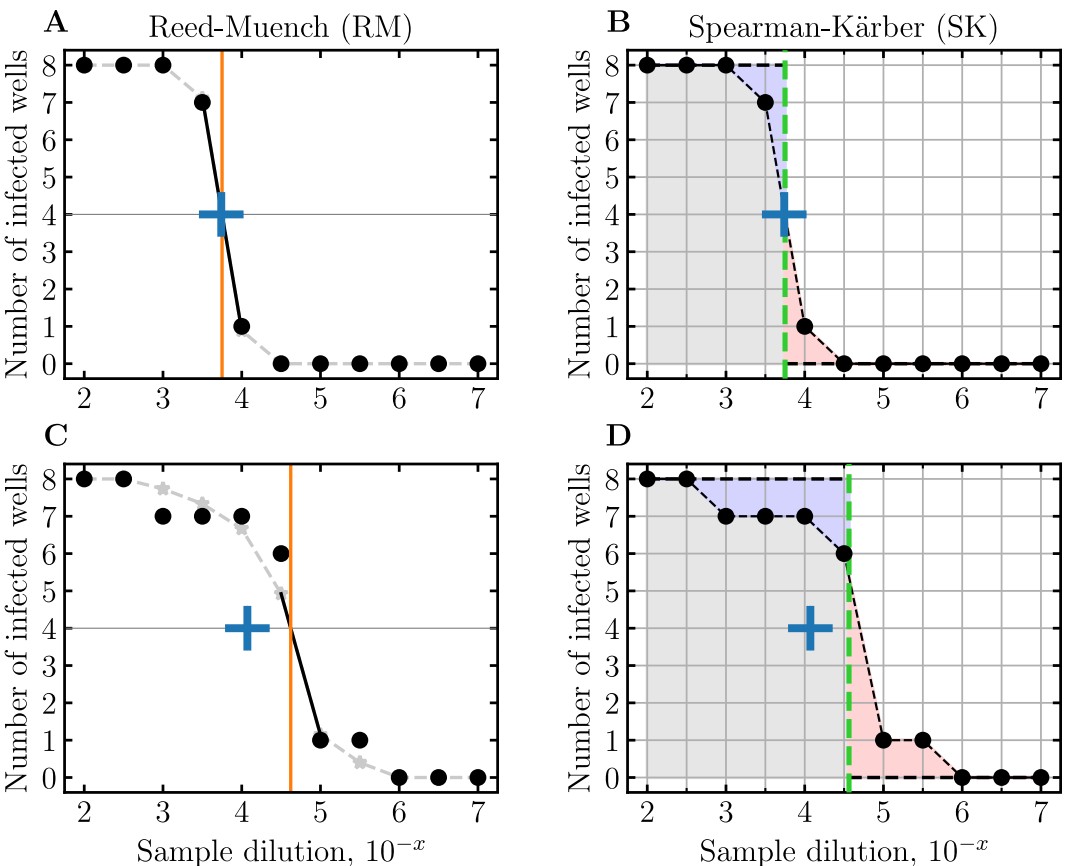

**Fig 4. Visualizing TCID$_{50}$ estimation by the RM and SK methods.** A,C: The RM method first smooths the data by taking the cumulative sum of the number of infected wells from the highest to the lowest dilution, and that of uninfected wells from the lowest to the highest dilution (grey dashed curve). It then identifies the dilution (vertical solid orange line) corresponding to the smooth curve's 50% crossing point (4/8 wells, horizontal grey line) based on the highest dilution with > 50% wells infected, and the lowest dilution with < 50% wells infected. B,D: The SK method identifies the dilution (vertical dashed green line) such that the area under the curve to its right (pale red) would exactly fill the area over the curve to its left (pale blue). The agreement between the true TCID$_{50}$ (blue plus) and the RM and SK estimates is good for the symmetric ED plate outcome in (A,B), but poor for the more irregular outcome in (C,D).

sometimes accurate (Fig 4A and 4B), their simplicity often leads to biased estimates (Fig 4C and 4D).

In contrast, **midSIN** is robust to these issues. Fig 5 demonstrates how **midSIN** can provide an estimate for the log$_{10}$(SIN/mL) in a sample using the number of infected wells at a single dilution, as long as at least one well is uninfected if all others are infected or vice-versa. This is because **midSIN** relies on Bayesian inference, i.e., when more than one column is available, it uses information from each column successively to revise and improve its estimate. This allows **midSIN** to correct for even large deviations from theoretical expectations, and thus improves its accuracy.

Fig 6 illustrates how well the **midSIN**, RM, and SK methods recover a known input sample concentration in simulated ED experiments, based on a plate layout consisting of 11 dilutions ($\mathcal{D}_1 = 10^{-2}$ to $\mathcal{D}_{11} = 10^{-8}$), a dilution factor of 1/4, and 8 replicates per dilutions. The infection concentration estimated by **midSIN** is in excellent agreement with the input concentration. For the RM and SK methods, which estimate the log$_{10}$(TCID$_{50}$/mL) rather than the log$_{10}$(SIN/mL), the agreement is generally poor due to the bias they introduce. Furthermore,

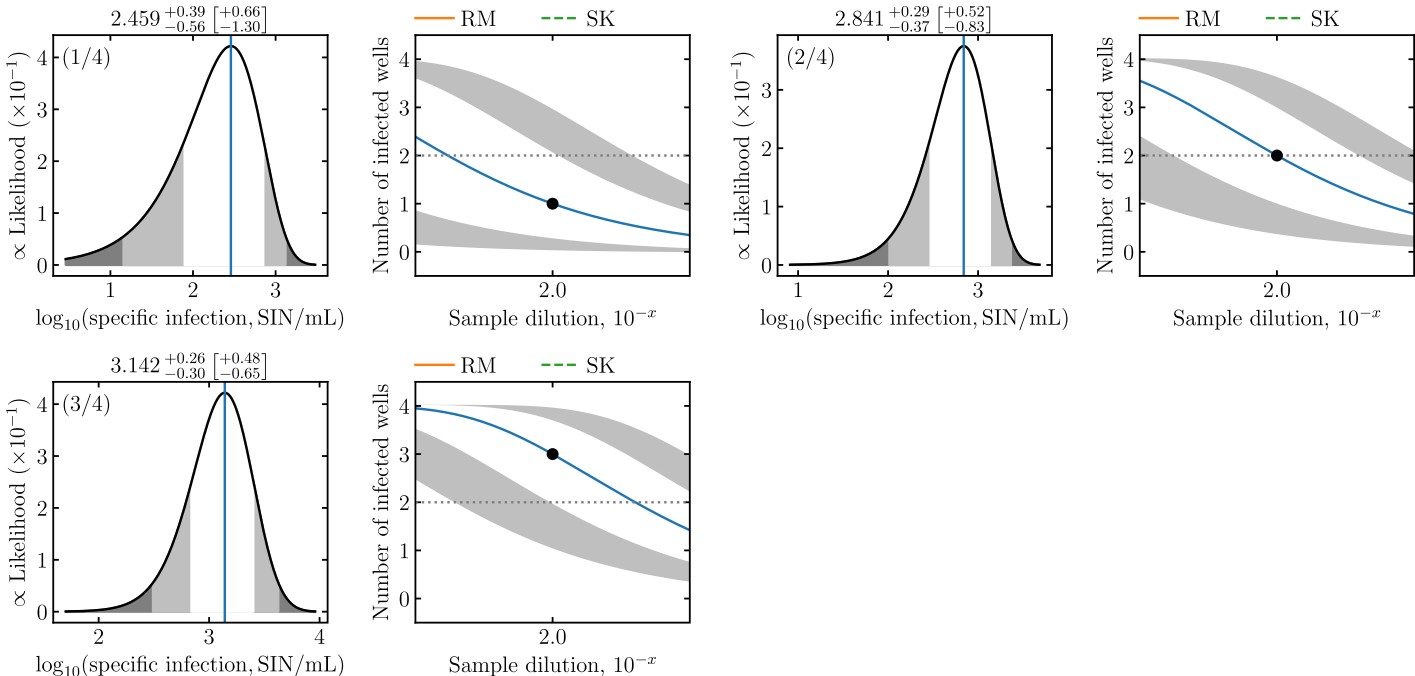

**Fig 5. midSIN's estimate of a sample's infection concentration based on a single dilution.** Simulated example of an ED plate with an inoculation volume of $V_{\text{inoc}} =$ 0.1 mL. Instead of serial dilutions, a single dilution ($\mathcal{D}_1 = 0.01$) is used, and either 1, 2 or 3 well(s) out of the 4 replicate wells are infected. As the fraction of infected wells increases, the uncertainty on the estimate (68% and 95% CIs) decreases, and the posterior distribution becomes more symmetric (Normal-like). Other features are as explained in the caption of Fig 1. The RM and SK methods cannot provide an estimate for these outcomes.

the RM and SK predictions are more variable (wavy pattern), and lose accuracy dramatically as the sample concentration approaches the limits of detection (the 2 ends) which, for the example plate layout simulated here, is around $10^3$ SIN/mL and $10^9$ SIN/mL. Interestingly, the basic calculations behind the RM and SK methods constrain the set of values they can return (sparsely populated grey histograms), compared to the more continuous range returned by **midSIN**, which contributes to its increased accuracy.

## Estimate accuracy as a function of plate layout

In Fig 2, we observed that even for large discrepancies between the expected (right panels, blue curve) and observed (right panels, black dots) ED assay outcome, the uncertainty (CI) of **mid-SIN**'s estimate remains relatively unchanged. This apparent robustness is because the uncertainty is primarily determined by the experimental design, namely the change in dilution between columns (dilution factor) and the number of replicate wells per dilution. Fig 7 explores the impact of varying either only the dilution factor, or only the number of replicates at each dilution, or varying one at the expense of the other by using a fixed number of wells (96 wells). When using **midSIN**, smaller changes in dilution (e.g., going from a dilution factor of 2.2/100 to 61/100) or more replicates per dilution (4 to 24) each improves the measure's accuracy (narrower CIs) by comparable amounts, but only when the total number of wells is allowed to increase to accommodate the change. When the total number of wells used is fixed, changing one at the expense of the other leaves the accuracy (CI) unchanged. This is somewhat also true for the $\log_{10}(\text{TCID}_{50})$ output concentration estimated by the RM and SK methods. However, at the smallest dilution factors (10/100 and 2.2/100), the bias introduced by the RM and SK methods becomes even larger and more unpredictable. For the input concentration

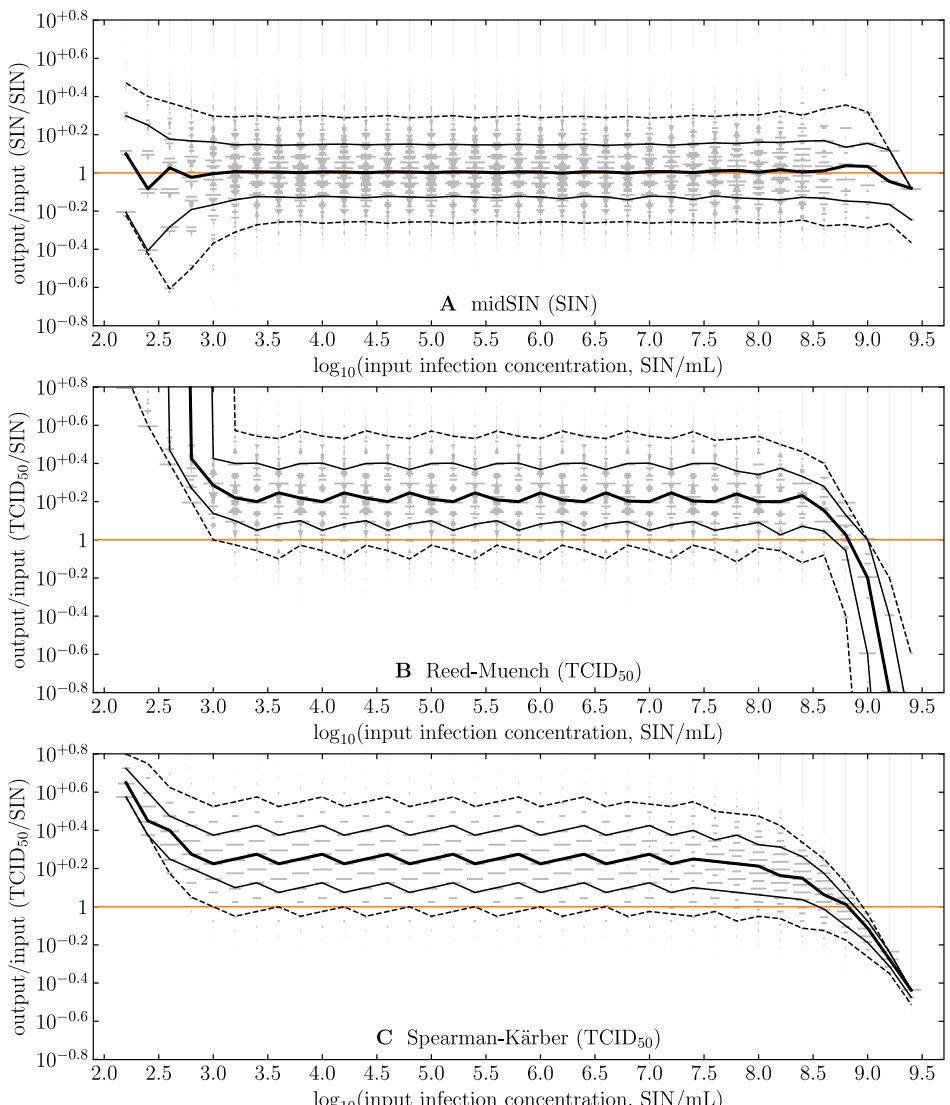

**Fig 6. Comparing known input to estimated output concentrations.** For each input concentration between $10^{2.2}$ and $10^{9.4}$, one million random ED experiment outcomes (# of positive wells in each dilution column) were generated. For each ED outcome, either A: **midSIN** was used to determine the most likely $\log_{10}(\text{SIN/mL})$; or the B: RM or C: SK method was used to estimate the $\log_{10}(\text{TCID}_{50}/\text{mL})$. Vertically stacked grey bands at each input concentration are sideways histograms, proportional to the number of ED outcomes that yield a given $y$-axis value. The black curves join the median (thick), 68$^{\text{th}}$ (thin) and 95$^{\text{th}}$ (dashed) percentile of the histograms, determined at (but not between) each input concentration. A plate layout of 11 dilutions, with 8 replicates per dilution, an inoculum volume of $V_{\text{inoc}} = 0.1$ mL, serial dilutions from $\mathcal{D}_1 = 10^{-2}$ to $\mathcal{D}_{11} = 10^{-8}$, separated by a dilution factor of $10^{-0.6} \approx 1/4$ were used in the simulated ED experiments.

considered in Fig 7 ($10^5$ SIN/mL), the dilution at which 50% of wells are infected is near the middle dilution. For sample concentrations such that 50% infected wells occur near or at the lowest or highest dilution chosen, the effect is even more significant.

Fig 7 also demonstrates that varying the dilution by smaller increments (e.g., a dilution factor of 61/100 rather than 10/100) provides greater granularity (uniqueness) of ED plate outcomes, and thus, greater accuracy of the $\log_{10}$ infection concentration estimates. Here, a distinct plate outcome means a distinct number of infected wells at each dilution, with no

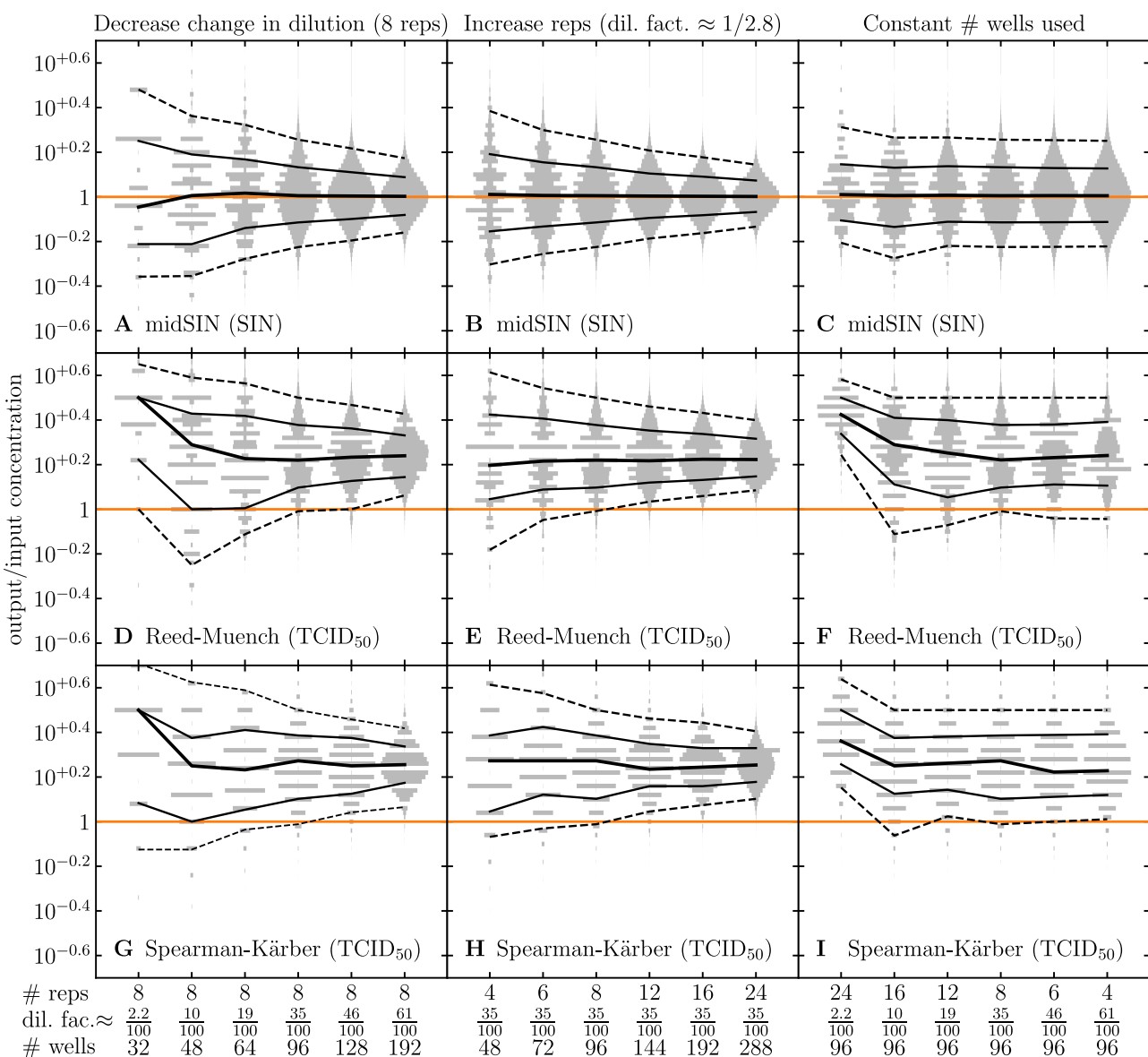

**Fig 7. Comparing the effect of the dilution factor and number of replicates per dilution.** The effect of either A,D,G: decreasing the change in dilution (from a dilution factor of 2.2/100 to 61/100) while keeping 8 replicates per dilution; or B,E,H: increasing the number of replicates per dilutions (4 to 24) while keeping a fixed dilution factor ($\approx 35/100$); or C,F,I: increasing the dilution factor while decreasing the number of replicates, keeping a fixed number of 96 wells used in total to titer one virus sample. Different rows represent the ratio of the estimated output concentration using (A–C) **midSIN** in SIN/mL, (D–F) RM or (G–I) SK in TCID$_{50}$/mL, and the input concentration. In all cases (A–I), the input concentration was $10^5$ SIN/mL, and as the dilution factor was varied, the highest and lowest dilutions in the simulated ED plate were held fixed to $\mathcal{D}_1 = 10^{-2}$ and $\mathcal{D}_{\text{last}} = 10^{-7}$, respectively, by changing the total # of dilutions performed (simulated). Everything else is generated, computed, and represented visually as described in the caption of Fig 6.

distinction as to exactly which of the replicate wells (e.g., the second versus the fourth) is infected at each dilution. An ED plate with serial dilutions ranging over 6 orders of magnitude (e.g., $10^{-2}$ to $10^{-7}$), with 4 different dilutions and 24 replicates/dilution (i.e., dilution factor of 2.2/100) provides $\sim 10^6$ ($[24 + 1]^4$) possible, distinct ED plate outcomes (Fig 7C, 7F and 7I, leftmost histogram). In contrast, a plate with the same serial dilution range, but with 24 different dilutions and 4 replicates/dilution (i.e., dilution factor of 61/100) yields $\sim 10^{17}$ ($[4 + 1]^{24}$)

distinct outcomes (Fig 7C, 7F and 7I, rightmost histogram). More generally, $[\text{reps} + 1]^{\text{dils}}$ is the number of distinct plate outcomes for a chosen number of dilutions (dils) and replicates (reps). Having fewer possible plate outcomes means that a larger range of concentrations would share the same most-likely ED plate outcome, yet each plate outcome only maps to one (the most likely) concentration estimate. This means that with fewer dilutions, the concentration estimate is forced to take on the nearest possible value it can take (Fig 7, the next closest grey band in the stack), and the accuracy of the concentration estimate is therefore reduced. So although having a greater number of dilutions is more labour intensive, it should be preferred over having a greater number of replicates per dilution.

## Discussion

We have introduced a new calculator tool called **midSIN** to replace the Reed-Muench (RM) and Spearman-Kärber (SK) calculations to quantify the infectivity of a virus sample based on an endpoint dilution (ED) assay. Rather than estimating the TCID$_{50}$ of a virus sample, **midSIN** calculates the number of infections the sample will cause, reported in units of specific infections (SIN). It does so without requiring any changes to current ED assay protocols, and can be accessed for free via an open-source web-application (https://midsin.physics.ryerson.ca). Importantly, because the SIN of a virus sample corresponds to the number of infections it will cause, it can be used directly to determine what dilution of the sample will achieve the desired multiplicity of infection (MOI).

Using a combination of *in vitro* and simulated experimental data, we demonstrated that **midSIN** provides more accurate and robust estimates than the biased RM and SK approximations. We confirmed that the RM and SK approximations overestimate the TCID$_{50}$ by 23.5%, such that 1 TCID$_{50}$ estimated by these methods will cause 1.781 rather than 1.44 infections [5, 6]. While, in theory, the intended MOI can be obtained by multiplying the TCID$_{50}$ by 0.7 (or rather $\ln(2) = 0.693$), one should instead multiply by 0.561 to account for the overestimation by RM and SK. Even when accounting for the overestimation, we showed that these methods perform particularly poorly when too few replicate wells per dilutions are used or when the change in dilution is large between successive serial dilutions. The two methods perform especially poorly when quantifying samples whose infection concentration approaches, but is still well within, the detection limit of the ED assay. In such cases, the bias introduced by these methods becomes even larger and more significant. For example, if the minimum and maximum dilutions of an ED plate are $10^{-2}$ and $10^{-8}$, virus samples with a concentration less than $10^{2.2}$ SIN or greater than $10^{7.6}$ SIN per inoculated well volume (typically 0.1 mL), will see their concentration estimated with an even larger bias by the RM and SK methods.

Using **midSIN** to measure the infectivity of a virus sample based on an ED assay does not require any change to ED experimental protocols and methods currently in use in one's laboratory (e.g., dilution factor, replicate per dilution, minimum dilution). Indeed, we demonstrated that **midSIN** can estimate a virus sample's SIN concentration based on even just a single dilution, as long as replicate wells at that dilution are not all infected or all uninfected. For a given number of ED wells used to titrate the sample and fixed minimum and maximum dilutions (ED detection range), we showed that having smaller changes between dilutions should be favoured over more replicates at each dilution. For example, using 11 dilutions, with a 4-fold dilution factor between dilutions and 8 replicate wells per dilution uses up 88 wells, leaving 8 wells of a 96-well plate for controls. This ED plate design, analyzed using **midSIN**, accurately measures virus sample concentrations ranging over $\sim 6$ orders of magnitude (e.g., $[10^1 – 10^7]$ SIN/mL, or $[10^6 – 10^{12}]$ SIN/mL, etc.) with an accuracy of $\sim 1.6$-fold ($\times 10^{\pm 0.2}$, 95% CI). In comparison, using 7 dilutions, with a 10-fold dilution factor, and 4 replicates (which

uses 28 rather than 88 wells) would also span 6 orders of magnitude, but with an accuracy of $\sim$3.2-fold ($\times 10^{\pm 0.5}$, 95% CI). To put these 2 accuracies in perspective: 1 mL of a sample measured to contain 10 SIN/mL, is expected to yield either 6–16 or 3–31 infections 95% of the time, given an accuracy of either $\times 10^{\pm 0.2}$ or $\times 10^{\pm 0.5}$ SIN/mL, respectively. Such an important decrease in accuracy means a reduced ability to detect experimental changes as statistically significant, with the $\times 10^{\pm 0.5}$ accuracy requiring a >10-fold change for statistical significance. Failing to identify a change as statistically significant as part of a study is far more costly than using more wells for each sample to increase measurement accuracy, and thus the statistical power of the study.

The **midSIN**-estimated SIN obtained from an ED assay was also compared to the PFU from a plaque assay for a set of influenza A virus samples. When the plaque and ED assays are performed as identically as possible (cell type, incubation time, etc.), as was the case here, 1 SIN $\approx$ 1 PFU. This demonstrates that indeed **midSIN**'s SIN is a measure of the number of infections a virus sample will cause. However, the plaque and focus forming assays have experimental limitations (time required, sensitivity of target cells to overlay, limited to viruses that cause CPE, subjectivity in counting plaques/foci, etc.) that cause many researchers to titrate virus using ED assays. Indeed **midSIN**'s SIN is a measure of the number of infections a virus sample will cause, and estimating the SIN concentration of a virus sample using data from ED assays is accessible, accurate, and predictive.

The work herein focused on the virus sample infectivity estimated from an unmodified ED assay. In principle, further improvements in accuracy could be achieved through the use of machine-automated scoring of infected wells using fluorescence intensity or colorimetry. Plate readers can be quite expensive, as are the consumable compounds they require, such as fluorescent antibodies, or antibodies loaded with compounds that can precipitate in the presence of another (colorimeter). In contrast, staining with crystal violet, trypan blue, etc. is an inexpensive and efficient way to identify the widespread cellular pathogenic effect of infection by a lytic virus, as are red blood cells to identify the presence of notable virus concentration in the supernatant of a well infected with a hemagglutination-capable virus. Since the aim of the ED assay is merely to establish whether or not infection occurred, the scoring of a well as having been infected or not, even when done visually, is likely less ambiguous. Therefore, in future work, it would be interesting to compare human- vs machine-scoring of wells to evaluate this step's contribution to the accuracy of the measure obtained.

Beyond the work presented herein, the development of **midSIN** will continue online as we implement new features and inputs for integration with various colorimetric and fluorescence instruments. The ease of use of **midSIN** and the greater usefulness and relevance of SIN as a measure of a virus sample's infectivity make them far superior to the TCID$_{50}$, and other ID$_{50}$ measures. We hope to see them adopted widely.

## Methods

### The mathematics of the dose-response assay

**Considering a single well.**   Consider a virus sample of volume $V_{\text{sample}}$ which contains an unknown concentration of infectious virions, $C_{\text{inf}}$, which we aim to determine. Drawing a small volume, $V_{\text{inoc}} < V_{\text{sample}}$, from the sample of volume $V_{\text{sample}}$, is analogous to drawing balls out of a bag containing green and yellow balls, and considering green balls a success, and yellow ones a failure. It is a series of Bernoulli trials where

$n = V_{\text{inoc}}/V_{\text{vir}}$ is the number of draws, i.e., the number of virion-size volumes ($V_{\text{vir}}$) drawn from the sample to form the inoculum volume ($V_{\text{inoc}}$), analogous to the number of balls drawn.

$k$ is the number of successes, i.e., the number of infectious virions drawn from the sample to form the inoculum, analogous to the number of green balls drawn.

$p$ is the probability of success, i.e., the fraction of virion-size volumes in the sample that are occupied by infectious virions, analogous to the probability of drawing a green ball.

The probability of success, $p$, is related to the concentration of infectious virus in the sample, $C_{\text{inf}}$, as

$$p = \frac{\text{Number of virions in sample}}{\text{Number of virion-size volumes in the sample}} = \frac{C_{\text{inf}} V_{\text{sample}}}{V_{\text{sample}}/V_{\text{vir}}} = C_{\text{inf}} V_{\text{vir}},$$

where $C_{\text{inf}}$ is the quantity we aim to estimate. Unlike the ball analogy where it is easy to count how many green balls $k$ were drawn, after having drawn $n$ virion-size volumes from the sample into our inoculum, we cannot count how many infectious virions were drawn into the inoculum. However, if this inoculum is deposited onto a susceptible cell culture, we can observe whether or not infection occurs, and this would indicate that the inoculum contained at least one or more infectious virions. Note that, as explained in the Introduction, even a productively infectious virion, i.e., one capable of completing the full virus replication from attachment to progeny release, might not result in a productive infection. As such, from hereon, $C_{\text{inf}}$ is used to designate the concentration of specific infections in the sample, which is smaller or equal to the concentration of infectious virions, i.e., measures the subset of the infectious virions.

Having deposited the inoculum into one well of the 96-well plate of our ED experiment, the likelihood that the well will *not* become infected, $q_{\text{noinf}}$, corresponds to the likelihood of having drawn $k = 0$ infectious virions (or rather, specific infections) out of the $n$ virion volumes that make up our inoculum, namely

$$\begin{aligned} q_{\text{noinf}} &= \text{Binomial}(k = 0 | n = V_{\text{inoc}}/V_{\text{vir}}, p = C_{\text{inf}} V_{\text{vir}}) \\ &= \frac{n!}{0!(n-0)!} p^0 (1-p)^{n-0} = (1-p)^n \\ q_{\text{noinf}} &= (1 - C_{\text{inf}} V_{\text{vir}})^{V_{\text{inoc}}/V_{\text{vir}}} \end{aligned} \quad (1)$$

where $q_{\text{noinf}}$ can be simplified by realizing that

$$\ln(1-x) \overset{|x|<1}{=} -x - \frac{x^2}{2} - \frac{x^3}{3} - \ldots \overset{|x|\ll1}{\approx} -x$$

$$\ln(q_{\text{noinf}}) = \frac{V_{\text{inoc}}}{V_{\text{vir}}} \ln(1 - C_{\text{inf}} V_{\text{vir}}) \approx \frac{V_{\text{inoc}}}{V_{\text{vir}}} (-C_{\text{inf}} V_{\text{vir}}) = -C_{\text{inf}} V_{\text{inoc}} .$$

As such,

$$q_{\text{noinf}} = (1 - C_{\text{inf}} V_{\text{vir}})^{V_{\text{inoc}}/V_{\text{vir}}} \approx \exp[-C_{\text{inf}} V_{\text{inoc}}] \quad (2)$$

where $q_{\text{noinf}}$ and $(C_{\text{inf}} V_{\text{vir}}) \in [0, 1]$ because $C_{\text{inf}} = N_{\text{vir}}/V_{\text{sample}}$ and the number of specific infections in the sample, $N_{\text{vir}}$, is at a minimum zero, and at most the maximum number of virion-size volumes that can physically fit in the sample volume, namely $V_{\text{sample}}/V_{\text{vir}}$. As such, the maximum possible infection concentration, given a sample of volume $V_{\text{sample}}$, is $C_{\text{inf}} = (V_{\text{sample}}/V_{\text{vir}})/V_{\text{sample}} = 1/V_{\text{vir}}$, and $C_{\text{inf}} \in [0, 1/V_{\text{vir}}]$.

**Considering replicate wells at a given dilution.** The ED assay is based on serial dilutions of the sample, with each dilution separated by a fixed dilution factor. We define the dilution factor $\in (0,1)$ as the fraction of the inoculum volume drawn from the previous dilution. For example, if the inoculum for a well, $V_{\text{inoc}} = 100$ μL, comprises 10 μL drawn from the previous dilution and 90 μL of dilution media, the dilution factor is $10/100 = 0.1$. If the serial dilution begins with a dilution of $\mathcal{D}_1 = 0.2$, then the following dilution will be $\mathcal{D}_2 = 0.02$. In Eq (1), the dilution under consideration, $\mathcal{D}_i$, will affect $n$, the number of virion-sized volumes drawn from the sample and deposited into the wells of the $i^{\text{th}}$ dilution, such that now $n = \mathcal{D}_i V_{\text{inoc}}/V_{\text{vir}}$. Therefore, the probability that a well at the $i^{\text{th}}$ dilution will *not* become infected is given by

$$q_i \equiv q_{\text{noinf}}^{\mathcal{D}_i} = \left(1 - C_{\text{inf}} V_{\text{vir}}\right)^{\mathcal{D}_i V_{\text{inoc}}/V_{\text{vir}}} \approx \exp[-C_{\text{inf}} V_{\text{inoc}} \mathcal{D}_i] \tag{3}$$

where $1 - q_i$ is the probability of infection for a well at the $i^{\text{th}}$ dilution, where $\mathcal{D}_i \in [0, 1]$.

When conducting an ED assay, each dilution in the assay contains a number of independent infection wells (replicates), all inoculated with the same dilution, $\mathcal{D}_i$. This is analogous again to drawing balls out of a bag, but this time there are $n_i$ draws (replicate wells), and the probability of success (i.e., that a well becomes infected) is simply one minus the probability of failure (i.e., that a well does not become infected, $q_i$). The probability that $k_i$ out of the $n_i$ wells become infected at dilution $\mathcal{D}_i$, is described by the Binomial distribution

$$\text{Binomial}(k = k_i | n = n_i, p = 1 - q_i) = \frac{n_i!}{k_i!(n_i - k_i)!} \left(1 - q_i\right)^{k_i} q_i^{n_i - k_i} \propto \left(1 - q_{\text{noinf}}^{\mathcal{D}_i}\right)^{k_i} q_{\text{noinf}}^{\mathcal{D}_i(n_i - k_i)}$$

where $n_i$ is the number of replicate wells at each dilution, but could be less if any well at dilution $\mathcal{D}_i$ are spoiled or contaminated.

However, our interest is not in determining $k_1$ given $q_{\text{noinf}}$, but rather in determining $q_{\text{noinf}}$ given that we observed $k_1$ infected wells out of $n_1$ wells in the first column. To this aim, we can make use of Bayes' theorem which, in our context, can be expressed as

$$\mathcal{P}(p | \text{data}) = \frac{\mathcal{P}(\text{data}|p)\ \mathcal{P}(p)}{\int_0^1 \mathcal{P}(\text{data}|p)\ \mathcal{P}(p)\ \mathrm{d}p}$$

or rather

$$
\begin{aligned}
\mathcal{P}_{\text{post},1}(q_{\text{noinf}} | k_1) &= \frac{\mathcal{P}(k_1 | q_{\text{noinf}})\ \mathcal{P}_{\text{prior}}(q_{\text{noinf}})}{\int_0^1 \mathcal{P}(k_1 | q_{\text{noinf}})\ \mathcal{P}_{\text{prior}}(q_{\text{noinf}})\ \mathrm{d}q_{\text{noinf}}} \\[2mm]
&= \frac{\left[(1 - q_{\text{noinf}}^{\mathcal{D}_1})^{k_1} q_{\text{noinf}}^{\mathcal{D}_1(n_1 - k_1)}\right] \mathcal{P}_{\text{prior}}(q_{\text{noinf}})}{\int_0^1 \mathcal{P}(k_1 | q_{\text{noinf}})\ \mathcal{P}(q_{\text{noinf}})\ \mathrm{d}q_{\text{noinf}}} \\[2mm]
\mathcal{P}_{\text{post},1}(q_{\text{noinf}} | k_1) &\propto \left[(1 - q_{\text{noinf}}^{\mathcal{D}_1})^{k_1} q_{\text{noinf}}^{\mathcal{D}_1(n_1 - k_1)}\right] \mathcal{P}_{\text{prior}}(q_{\text{noinf}})
\end{aligned}
$$

where $\mathcal{P}_{\text{post},1}(q_{\text{noinf}} | k_1)$ is our updated, posterior belief about $q_{\text{noinf}}$ after having observed $k_1$ successes out of $n_1$ trials in the first column ($i = 1$), and given our prior belief, $\mathcal{P}_{\text{prior}}(q_{\text{noinf}})$, about $q_{\text{noinf}}$ before making this observation.

**Considering all dilutions of the ED assay.** As mentioned above, in the 96-well ED assay, each dilution contains a number of independent infection wells (replicates) inoculated with the same sample concentration. This process is then repeated over a series of dilutions, each separated from the previous by a fixed dilution factor. Having observed the fraction of wells infected at the first dilution considered, $\mathcal{D}_1$, we have updated our posterior belief about $q_{\text{noinf}}$. We will now use this updated belief as our new prior as we observe our second dilution ($\mathcal{D}_2$),

such that

$$
\begin{aligned}
\mathcal{P}_{\text{post,2}}(q_{\text{noinf}}|\vec{k}_2) &\propto \mathcal{P}(k_2|q_{\text{noinf}})\ \mathcal{P}_{\text{post,1}}(q_{\text{noinf}}|k_1)\\
\mathcal{P}_{\text{post,2}}(q_{\text{noinf}}|\vec{k}_2) &\propto [(1-q_{\text{noinf}}^{D_2})^{k_2}\, q_{\text{noinf}}^{D_2(n_2-k_2)}]\,[(1-q_{\text{noinf}}^{\mathcal{D}_1})^{k_1}\, q_{\text{noinf}}^{\mathcal{D}_1(n_1-k_1)}]\mathcal{P}_{\text{prior}}(q_{\text{noinf}})\\
\mathcal{P}_{\text{post,2}}(q_{\text{noinf}}|\vec{k}_2) &\propto \mathcal{Q}(\vec{k}_2|q_{\text{noinf}})\,\mathcal{P}_{\text{prior}}(q_{\text{noinf}})\ ,
\end{aligned}
$$

where we introduce $\vec{k}_2 = \{k_1, k_2\}$ and

$$
\mathcal{Q}(\vec{k}_2|q_{\text{noinf}}) = [(1-q_{\text{noinf}}^{D_2})^{k_2}\, q_{\text{noinf}}^{D_2(n_2-k_2)}]\,[(1-q_{\text{noinf}}^{\mathcal{D}_1})^{k_1}\, q_{\text{noinf}}^{\mathcal{D}_1(n_1-k_1)}]
$$

as short-hands for convenience. From this, it is easy to extrapolate the posterior distribution after having observed all $J$ dilutions ($\mathcal{D}_1, \mathcal{D}_2, \ldots, \mathcal{D}_J$) of the ED assay, namely

$$
\mathcal{P}_{\text{post,J}}(q_{\text{noinf}}|\vec{k}_J) \propto \mathcal{Q}(\vec{k}_J|q_{\text{noinf}})\,\mathcal{P}_{\text{prior}}(q_{\text{noinf}}) \tag{4}
$$

where

$$
\mathcal{Q}(\vec{k}_J|q_{\text{noinf}}) = \left[\prod_{j=1}^{J}(1-q_{\text{noinf}}^{\mathcal{D}_j})^{k_j}\right] q_{\text{noinf}}^{\sum_{j=1}^{J}\mathcal{D}_j(n_j-k_j)}\ . \tag{5}
$$

Note that this expression is largely equivalent to that obtained by Mistry et al. [8] in the context of estimating the TCID$_{50}$ of a virus sample, and by many others in the broader context of infection dose quantification [12, 13].

**Considering the choice of prior.** In Eq (4), we obtained a posterior for $q_{\text{noinf}}$. Our objective, however, is to estimate the posterior distribution for $C_{\text{inf}}$, the specific infection concentration in our sample, rather than $q_{\text{noinf}}$. In fact, because both the plaque and ED assays provide an accuracy that is normally distributed in $\log_{10}(C_{\text{inf}})$ rather than $C_{\text{inf}}$, it follows that $\log_{10}(C_{\text{inf}})$ (hereafter $\ell_{\text{Cinf}}$) rather than $C_{\text{inf}}$ is the quantity of interest. We note that $\mathcal{Q}(\vec{k}_J|q_{\text{noinf}})$ in Eq (4) is a probability density function in $\vec{k}_J = \{k_1, k_2, \ldots, k_J\}$, rather than in $q_{\text{noinf}}$. As such, a change of variables from $q_{\text{noinf}}$ to $\ell_{\text{Cinf}}$ would affect only the prior, because $\mathcal{Q}(\vec{k}_J|q_{\text{noinf}}) = \mathcal{Q}(\vec{k}_J|q_{\text{noinf}}(\ell_{\text{Cinf}})) = \mathcal{Q}(\vec{k}_J|\ell_{\text{Cinf}})$. Thus, the posterior distribution for $\ell_{\text{Cinf}}$ is given by

$$
\mathcal{P}_{\text{post,J}}(\ell_{\text{Cinf}}|\vec{k}_J) \propto \mathcal{Q}(\vec{k}_J|q_{\text{noinf}}(\ell_{\text{Cinf}}))\ \mathcal{P}_{\text{prior}}(\ell_{\text{Cinf}})\ . \tag{6}
$$

To complete this expression, we need to choose a physically and biologically appropriate prior belief regarding $\ell_{\text{Cinf}}$. Prior to conducting the ED assay, we know at least that $C_{\text{inf}} \in [1/V_{\text{Earth}}, 1/V_{\text{vir}}]$, where $1/V_{\text{vir}}$ is the maximum possible concentration, namely that if the entire volume of the sample is constituted solely of infectious virions, and $1/V_{\text{Earth}}$ is the minimum possible concentration, namely that if there was only one infectious virion left on Earth. As we explain below, these limits are not important; only the fact that they are convincingly physically bounded both from above and below, i.e., $\in (0, \infty)$, is relevant.

If we choose our prior to be uniform in $C_{\text{inf}} \in [1/V_{\text{Earth}}, 1/V_{\text{vir}}]$, namely $\mathcal{P}_{\text{prior}}(C_{\text{inf}}) = 1/(1/V_{\text{vir}} - 1/V_{\text{Earth}}) \approx V_{\text{vir}}$, and using the fact that $\mathcal{P}_{\text{prior}}(C_{\text{inf}})\,\mathrm{d}C_{\text{inf}} = \mathcal{P}_{\text{prior}}(\ell_{\text{Cinf}})\,\mathrm{d}\ell_{\text{Cinf}}$, we can write

$$
\mathcal{P}_{\text{prior}}(\ell_{\text{Cinf}}) = \mathcal{P}_{\text{prior}}(C_{\text{inf}})\frac{\mathrm{d}C_{\text{inf}}}{\mathrm{d}\ell_{\text{Cinf}}} = V_{\text{vir}}\frac{\mathrm{d}[10^{\ell_{\text{Cinf}}}]}{\mathrm{d}\ell_{\text{Cinf}}} = V_{\text{vir}}\ \ln(10)10^{\ell_{\text{Cinf}}} \propto 10^{\ell_{\text{Cinf}}}
$$

which yields

$$\mathcal{P}_{\text{post},J}(\ell_{\text{Cinf}}|\vec{k}_J) \propto \mathcal{Q}(\vec{k}_J|q_{\text{noinf}}(\ell_{\text{Cinf}}))\ 10^{\ell_{\text{Cinf}}}\ . \tag{7}$$

We see here that the range chosen for the uniform prior in $C_{\text{inf}}$ is not important because it only contributes a constant to our proportionality Eq (6).

Alternatively, because the ED assay estimates $\ell_{\text{Cinf}}$ rather than $C_{\text{inf}}$, our prior belief about the virus concentration is more appropriately expressed in $\ell_{\text{Cinf}}$ rather than $C_{\text{inf}}$. Again, the bounds of the uniform distribution in $\ell_{\text{Cinf}}$ is unimportant, provided that it is finite in extent such that $\ell_{\text{Cinf}} \in \left[\ell_{\text{Cinf}_{\min}}, \log_{10}(1/V_{\text{vir}})\right]$ where $\ell_{\text{Cinf}_{\min}} > -\infty$, such that we can write

$$\mathcal{P}_{\text{post},J}(\ell_{\text{Cinf}}|\vec{k}_J) \propto \mathcal{Q}(\vec{k}_J|q_{\text{noinf}}(\ell_{\text{Cinf}}))\ . \tag{8}$$

Fig 8 illustrates the two distinct priors assumed to arrive at Eqs (7) and (8) and their impact on the posterior $\mathcal{P}_{\text{post},J}(\ell_{\text{Cinf}}|\vec{k}_J)$ for the example ED experiment described in Fig 1. Fig 8A illustrates the consequence of choosing a prior uniform in $C_{\text{inf}}$, i.e., a bias towards higher virus concentrations. This is because a uniform prior in $C_{\text{inf}}$ corresponds to a belief that one is as likely to measure a set of virus concentrations in the range [0.001, 0.002] as in the range [1,000, 000.001, 1, 000, 000.002]. When plotted on a log-scale, there are 100× more intervals of width 0.001 in $[10^4, 10^5]$ than in $[10^2, 10^3]$. Thus, this prior corresponds to a belief that the likelihood of measuring a certain virus concentration increases exponentially as $\ell_{\text{Cinf}}$ increases linearly. In contrast, a prior uniform in $\ell_{\text{Cinf}}$ corresponds to a belief that one is as likely to measure a set of virus concentrations in the range [0.001, 0.002] as in the range [1, 000, 000, 2, 000, 000], or rather in the range $[1, 2] \times 10^{-3}$ as in the range $[1, 2] \times 10^6$. As such, a uniform distribution in $\ell_{\text{Cinf}}$ is more physically and biologically sensible and therefore was chosen for our estimation method.

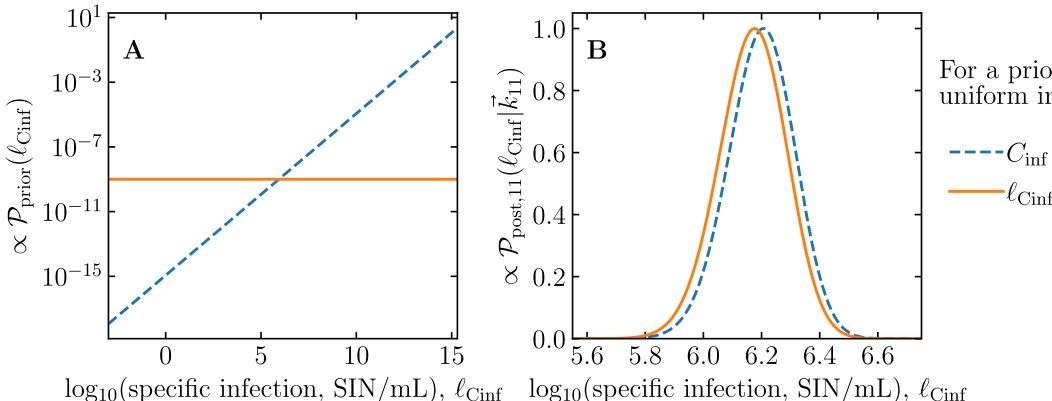

**Fig 8. Impact of the choice of prior on the posterior distribution for $\ell_{\text{Cinf}}$.** A: Non-normalized priors for $\log_{10}$(specific infections, SIN/mL) = $\ell_{\text{Cinf}}$ that are uniform in either $C_{\text{inf}}$ or $\ell_{\text{Cinf}}$ are shown. A prior uniform in $C_{\text{inf}}$ is biased towards larger values of $\ell_{\text{Cinf}}$. B: Updated posterior belief about $\ell_{\text{Cinf}}$ for each of the two prior beliefs shown in A, as per Eqs (7) and (8), after having observed the ED assay example provided in Fig 1. While the prior uniform in $C_{\text{inf}}$ yields a posterior with a mode of $\ell_{\text{Cinf}} = 6.21$, that for a prior uniform in $\ell_{\text{Cinf}}$ yields a mode of $\ell_{\text{Cinf}} = 6.18$.

## Calculation of midSIN's outputs

One of the graphical outputs of **midSIN** is the non-normalized posterior distribution of $\ell_{\text{Cinf}}$ given the number of wells that were infected at each dilution, $\vec{k}_J$, like that shown in Fig 1(left panel), computed as

$$\mathcal{U}_{\text{post}}(\ell_{\text{Cinf}} | \vec{k}_J) = \prod_{j=1}^{J} \frac{n_j!}{k_j! \, (n_j - k_j)!} \cdot p_j^{k_j} \cdot (1 - p_j)^{n_j - k_j} \tag{9}$$

where

$$p_j = 1 - \exp[-10^{\ell_{\text{Cinf}}} \cdot V_{\text{inoc}} \cdot \mathcal{D}_j] . \tag{10}$$

While $\mathcal{U}_{\text{post}}$ is not the normalized posterior distribution for $\ell_{\text{Cinf}}$, its maximum value at its mode ($\ell_{\text{Cinf,mode}}$) is the normalized probability of observing this particular ED plate outcome ($\vec{k}_J$) out of all other possible plate outcomes, assuming the true, specific infection concentration in the sample is $\ell_{\text{Cinf,mode}}$.

Another visual output of **midSIN** is a graphical representation of the theoretical number of wells that would be infected given the most likely $\ell_{\text{Cinf}}$, like that shown in Fig 1(right panel). It is computed following

$$N_{\text{wells infected}}(x) = N_{\text{wells total}}[1 - \exp(-10^{\ell_{\text{Cinf,mode}}} V_{\text{inoc}} 10^{-x})] , \tag{11}$$

where $x$ is the $\log_{10}$ of the dilution such that $\mathcal{D} = 10^{-x}$ is the dilution. It corresponds to the continuous equivalent of this quantity which is discrete in the ED assay, namely $\mathcal{D}_i = 10^{-x_i}$ which is the $i^{\text{th}}$ dilution of the sample. As such, $\mathcal{D}_i = (\text{minimum dilution}) \cdot (\text{dilution factor between columns})^{i-1}$ where $i \in [1, J]$. For example, if the dilution of the least diluted column is $0.1 = 10^{-1}$ and the dilution factor between dilutions in the ED assay is such that it halves the concentration between each dilution, i.e., $1/2 = 2^{-1} = 10^{-\log_{10}(2)} \approx 10^{-0.301}$, then $\mathcal{D}_i = 10^{-1} \cdot 10^{-0.301 \cdot (i-1)}$ such that $\mathcal{D}_1 = 10^{-1}$, $\mathcal{D}_2 = 10^{-1.301}$, $\mathcal{D}_3 = 10^{-1.602}$, and so on, such that $x_1 = 1$, $x_2 = 1.301$, $x_3 = 1.602$, and so on.

In the graphical representation of the ED assay, the edges of the grey bands flanking the theoretical blue curve correspond to Eq (11) wherein $\ell_{\text{Cinf,mode}}$ has been replaced by the 68% and 95% CI values for $\ell_{\text{Cinf}}$. These CI bands *do not* correspond to the 68% and 95% CI of the expected number of infected wells at each dilution given $\ell_{\text{Cinf,mode}}$.

The sample dilution corresponding to 1 TCID$_{50}$ estimated based on the biased RM and SK approximations (right panels) are converted to SIN (left panels) based on 1 TCID$_{50}$ = $e^{\gamma = 0.5772}$ SIN = 1.781 SIN [5, 6]. In contrast, the $\log_{10}(\text{SIN/mL})$ computed by **midSIN** can be converted to a true (unbiased) estimate of $\log_{10}(\text{TCID}_{50})$ using 1 TCID$_{50}$ = $1/\ln(2)$ SIN = 1.44 SIN [4].

## Infection concentration measures of influenza A virus samples

**Cell culture.** Madin-Darby canine kidney cells (MDCKs) were cultured in growth media (complete MEM media with 5% heat-inactivated FBS), in tissue culture treated T75 flasks, at 37°C with 5% CO$_2$ and 95% relative humidity. Cells were split 1/10 every 3–4 days or upon reaching approximately 95% confluency. One passage of cells was expanded for use by both researchers in one experiment to quantify the 50% tissue culture infectious dose (TCID$_{50}$) and plaque forming units (PFU) of one viral strain.

**Viral stocks.** Stocks of influenza A/Puerto Rico/8/34 (H1N1) (PR8) and influenza A/California/4/09 (Cali09) were stored at -80°C and thawed on ice immediately before use. The

TCID$_{50}$ and PFU of stock viruses was known to both researchers prior to this study. Serial dilutions were made in MDCK infection media (complete MEM media with 4.25% BSA) and dilutions were made by each researcher independently for titering. 'Researcher A' and 'Researcher B' independently performed the TCID$_{50}$ and PFU assays of one viral strain for one experiment on the same day using the same viral stock, reagents, and passage of cells. Each experiment was performed on a separate day (Fig 3).

**Plaque assay.**   MDCKs were seeded in six-well plates ($5.5 \times 10^5$ cells/mL, 2 mL/well) and grown to 90% confluency overnight (37˚C, 5% CO$_2$, 95% relative humidity). Each six-well plate contained 10-fold serial dilutions plated in singlet as well as a negative control and five 6-well plates were carried out per experiment. Cells were washed twice with PBS containing Ca$^{2+}$Mg$^{2+}$ (PBS w/ Ca$^{2+}$Mg$^{2+}$) (Gibco), before the addition of 500 µL of viral dilutions per well. After 1 h at room temperature on a rocker, the inoculum was aspirated, cells were washed with PBS w/ Ca$^{2+}$Mg$^{2+}$, and gently covered with 2 mL of agarose overlay (complete media, 4.25% BSA, 0.9% agarose, 1 µg/mL TPCK-Trypsin). After drying the overlay at room temperature, plates were inverted and incubated (37˚C, 5% CO$_2$, 95% relative humidity) for 3 d (PR8) or 4 d (Cali/09). Plaques were visualized by staining cells with 0.1% crystal violet solution in 37% formaldehyde for 30 min and counted by 'Researcher A' or 'Researcher B' on their respective experiments (Fig 3).

**TCID$_{50}$ assay.**   MDCKs were seeded in 96-well flat bottom plates ($5 \times 10^4$ cells/100 µL, 100 µL/well) and grown to 80% confluency overnight (37˚C, 5% CO$_2$, 95% relative humidity). For each experiment, 4 replicate wells, at each of 7 different dilutions separated by a 10-fold dilution, were infected, and the dilution series was performed 5 times. Cells were washed with PBS w/ Ca$^{2+}$Mg$^{2+}$ before the addition of 100 µL of viral dilutions per well. After 1 h at room temperature on a rocker, the inoculum was aspirated and replaced with 100 µL of infection media containing 1 µg/mL TPCK-Trypsin. Cells were incubated (37˚C, 5% CO$_2$, 95% relative humidity) for 3 d (PR8) or 4 d (Cali/09). Supernatants from each of the MDCK-containing wells were transferred to a matching well in a 96-well U-bottom plate in the same configuration, and mixed with chicken red blood cells (30 min, room temperature). This enabled us to score each of the original MDCK-containing wells as either positive or negative for infection, based on whether their supernatant caused hemagglutination. This was performed and read by 'Researcher A' or 'Researcher B' on their respective experiments.

**Statistical analysis.**   The data points reported in Fig 3C and 3D were computed by taking each of the 5 replicates measured with either the PFU, RM, or SK and the 5 replicates measured via SIN (5 replicates × 5 replicates = 25 pairs) for each of the 2 experiments by each of the 2 researchers, yielding 100 pairs. For each pair, the log$_{10}$ of ratio of either PFU, RM or SK over SIN was computed. The mean and standard deviation of the resulting 100 log$_{10}$(ratio) were computed and are reported in Fig 3C and 3D. The statistical significance ($p$-value) of the differences between (PFU,RM,SK) and (SIN) was computed using the Mann-Whitney U test (`scipy.stats.mannwhitneyu`).

# Author Contributions

**Conceptualization:** Donald C. Warren, Amanda P. Smith, Catherine A. A. Beauchemin.

**Data curation:** Daniel Cresta, Donald C. Warren, Amanda P. Smith, Lindey C. Lane, Amber M. Smith, Catherine A. A. Beauchemin.

**Formal analysis:** Daniel Cresta, Donald C. Warren, Catherine A. A. Beauchemin.

**Funding acquisition:** Amber M. Smith, Catherine A. A. Beauchemin.

**Investigation:** Daniel Cresta, Donald C. Warren, Christian Quirouette, Amanda P. Smith, Lindey C. Lane, Amber M. Smith, Catherine A. A. Beauchemin.

**Methodology:** Daniel Cresta, Donald C. Warren, Christian Quirouette, Amanda P. Smith, Lindey C. Lane, Amber M. Smith, Catherine A. A. Beauchemin.

**Project administration:** Amber M. Smith, Catherine A. A. Beauchemin.

**Resources:** Christian Quirouette, Catherine A. A. Beauchemin.

**Software:** Daniel Cresta, Christian Quirouette, Catherine A. A. Beauchemin.

**Supervision:** Amber M. Smith, Catherine A. A. Beauchemin.

**Validation:** Daniel Cresta, Donald C. Warren, Christian Quirouette, Catherine A. A. Beauchemin.

**Visualization:** Daniel Cresta, Donald C. Warren, Christian Quirouette, Catherine A. A. Beauchemin.

**Writing – original draft:** Daniel Cresta, Catherine A. A. Beauchemin.

**Writing – review & editing:** Daniel Cresta, Donald C. Warren, Christian Quirouette, Amanda P. Smith, Lindey C. Lane, Amber M. Smith, Catherine A. A. Beauchemin.

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
