## [Decision Letter · Decision Letter 0]

2 May 2021

Dear Dr. Beauchemin,

Thank you very much for submitting your manuscript "Time to revisit the endpoint dilution assay and to replace TCID50 and PFU as measures of a virus sample's infection concentration" for consideration at PLOS Computational Biology.

As with all papers reviewed by the journal, your manuscript was reviewed by members of the editorial board and by several independent reviewers. In light of the reviews (below this email), we would like to invite the resubmission of a significantly-revised version that takes into account the reviewers' comments.

Your paper was reviewed by an expert on the statistics of estimating ID50s, and two experimental virologists with a strong quantitative research agenda. The two experimental reviewers saw great value in the approach you are proposing. All reviewers agree that the paper could be improved by shortening it, focusing on describing your method. The more general parts on the various benefits or drawbacks of plaque-forming and dilutions assays could be skipped, especially because the experimentalists among the reviewers did not fully agree with the material presented. We also believe that your paper would improve by comparing your method to more recent advances in the estimation of TCID50 as Reed and Munch is outdated (although still used in some circles). Reviewer 1 gives a few starting references for a more comprehensive comparison with existing methods.

We cannot make any decision about publication until we have seen the revised manuscript and your response to the reviewers' comments. Your revised manuscript is also likely to be sent to reviewers for further evaluation.

Sincerely,

Roland R Regoes

Associate Editor

PLOS Computational Biology

Rob De Boer

Deputy Editor

PLOS Computational Biology

Reviewer's Responses to Questions

**Comments to the Authors:**

Reviewer #1: The article uses Bayesian probabilities to calculate the posterior probability distribution of the effective concentration of infectious particles in a viral stock. It also implements the calculation on the web. The article is unsuitable for publication in its present form. The authors should announce the software (e.g., Google “journal bioinformatics software”), confining their explanation to a few pages at most. The article omits an adequate survey of even rudimentary references. The end of the review lists some references relevant to similar problems in animal trials, not necessarily for citation, but as a start for searching for appropriate citations.

Most of the authors’ explanation is unnecessary. The discussion of the relative merits of plaque-forming and dilutions assays, e.g., is irrelevant. Each type of assay has its merits and drawbacks, but the decision to use one or the other is subordinate to experimental means and ends. The article can therefore take the use of a dilution assay as dependent on ends, as a given. The article motivates itself with the Spearman-Karber and Reed-Muench methods. Although the methods still appear in the literature, they have been discredited for at least 30 years. The article’s notation also obscures the simplicity of its ideas. Psychological experiments have shown that mathematical subscripts should be single letters, preferably with mnemonic value, because lengthy subscripts slow readers’ comprehension. To appreciate the point, replace q[noinf] by q (without subscript) in all equations.

The Bayesian probability model motivating the article is routine. Physically, infection is modeled by a Poisson likelihood. The article then gives a lengthy physical justification of the model prior. A routine non-informative prior may be preferable, but in any case, a Bayesian posterior should not be sensitive to the prior but depend mostly on the data. Any lengthy physical justification of the prior is therefore irrelevant.

Here are the promised citations.

Calculation of ID-50 https://pubmed.ncbi.nlm.nih.gov/26285041/

Infectious Particle Concentration https://pubmed.ncbi.nlm.nih.gov/1323844/

Harold Jeffreys "Theory of Probability" discusses non-informative priors, placing in context the unimportance of a "physical" Bayesian prior in statistical calculations.

Reviewer #2: This paper describes a new and better method for computing viral titers from endpoint dilution assays. The method is accompanied by an online and downloadable calculator. As a virologist whose lab routinely performs TCID50s, I can say that the method used here is a substantial improvement, and is something I am going to start recommending to my group. I definitely think that based on its content, this paper deserves to be published in PLoS Computational Biology.

That said, I have some pretty substantive suggested revisions. One of these is something that I think needs to be removed, and the others are my impression of things that need to be changed if the paper is going to be understood and well received by virologists.

MAJOR COMMENTS:

The paper makes two arguments: (1) endpoint dilutions assays such as TCID50 are better than plaque assays, and (2) the midSIN method is better than things like Reed-Muench for computing titers from endpoint dilution assays. The second of these points is definitely true, and forms the strong basis for the content of this paper. However, I don’t think the first point (superiority of endpoint dilution over plaque assay) is clearly established, nor do I think it’s at all necessary for this paper. I say this as someone who personally prefers endpoint dilution assays (TCID50) to plaque assay. But some virologists prefer plaque assays for a variety of reasons, including liking to see the plaques, the additional information they get from examining plaque sizes, etc. If I hadn’t read all the way through because I was a reviewer, I would have dismissed this paper after the first few paragraphs as an opinion piece arguing for TCID50 over plaque assays, and not paid attention to any of the rest. I strongly recommend the authors focus on what they clearly objectively demonstrate (that the midSIN method is better than alternatives for computing endpoint titers), and dispense with the more subjective arguments based on experimental factors that make them personally prefer endpoint assays to plaque assays.

I think the paper would benefit from a clearer “intuitive layman’s explanation” of what exactly is wrong with the Reed-Muench formula compared to midSIN. Right now there is little explanation in main text, and then highly technical details in Methods but not good bridging of these.

Although this is more a stylistic comment and one that is ultimately at the authors’ discretion, I’d suggest that the paper will have more impact if it’s more succinct, has less vague discussion of experiments and philosophical issues of titers, and really cuts more quickly to the heart of the issue which is that they have an improved way to calculate endpoint titers, they have implemented a calculator, and that their method allows calculation of how experimental choices (like dilution factor, number of dilution series, etc) affect accuracy.

MINOR COMMENTS:

The number of acronyms introduced just in the abstract (RM, SK, etc) becomes overwhelming and decreases readability. Maybe some of the less commonly used acronyms could be eliminated in favor of just writing out the full phrase?

Lines 6-17: another limitation of counting virions under a microscope is that it does not distinguish physical from infectious particles. The same is true for qPCR. This is a really serious limitation, moreso even than cost, etc. In fact, I sort of wonder if this entire first paragraph is a little bit irrelevant to the question at hand, which is titrating infectious particles.

Lines 19-22: Again, this isn’t quite true. They are certainly not easy to separate, but for instance with influenza there is some evidence that defective virions lacking genes sometimes have slightly different morphologies, etc—and can at least be partially separated by certain types of centrifugation. Again, like for lines 6-17, I sort of feel like the authors are spending a lot of time on not 100% accurate text that isn’t even really relevant to their main point and finding, which is titrating infectious particles.

Lines 68-71: The same limitation can apply to endpoint dilution (e.g., TCID50) assays, as the actual cell being used for the experiment doesn’t always work for the endpoint dilution assay. For instance, people performing flu infections of human primary airway cells still titer the virus by TCID50 on MDCK cells as you can’t do a TCID50 in human primary airway cells.

Reviewer #3: The manuscript by Cresta et al suggest a new platform for analyzing endpoint dilution assays of virus infection, which are currently based on mathematical approximations that introduce a bias into the outcome. While these errors are known and can be corrected, the calculations still show inconsistencies is specific cases.

I think its in general a good idea to revisit such traditional assays, identify potential limitations and work on their improvement. The authors also provide their analysis as an open online tool, which is a nice example for the open science principles. The manuscript is well written but the real value of the tool needs to be worked out a bit more. While the presented tool overcomes some limitations, its use seems overstated as it does not solve all of the mentioned problems.

1) There are many ways to quantify a virus sample and its important to consider what the assays measure. The authors aim at improving the estimation of an infection concentration meaning how many infections a virus sample could cause per unit volume. They compare plaque/focus forming assays with endpoint dilution methods. While the introduction gives a nice overview of these assays and highlights their limitations, in particular those of the plaque assay, it’s not a fair comparison. The plaque or focus assays use an overlay medium to restrict infection spread to only the neighboring cells. Infected cells and their infected neighbors will then after some time and following some form of coloring be visible as a plaque/focus. The ED assay (TCID50) typically does not use overlay medium and relies on immunostaining or CPE to score infection. Plaque and ED assay thus have usually different readouts, replication or infection. In the paper, the authors use hemagglutinating units to quantify the amount of released virus in their TCID50 assays. That is yet a different measure and might in include non-replicating particles.

In my opinion is the proposed midSIN platform best suited to analyze traditional ED assays where infection is labelled with antibodies and can be analyzed with an automated reader. The manuscript (title, intro and discussion) is thus misleading in several places by saying that midSIN overcomes the limitations of a plaque assay.

2) Another aspect that I think could be improved is the requirement of a threshold to score a well as positive. Can the analysis be performed using raw plate analyzer readings (fluorescence units per well)? That would be ideal and remove all personal bias from the analysis.

**Have the authors made all data and (if applicable) computational code underlying the findings in their manuscript fully available?**

Reviewer #1: Yes

Reviewer #2: Yes

Reviewer #3: Yes

PLOS authors have the option to publish the peer review history of their article (what does this mean?). If published, this will include your full peer review and any attached files.

Reviewer #1: No

Reviewer #2: No

Reviewer #3: No
---

## [Decision Letter · Decision Letter 1]

26 Sep 2021

Dear Dr. Beauchemin,

We are pleased to inform you that your manuscript 'Time to revisit the endpoint dilution assay and to replace TCID50 as a measure of a virus sample's infection concentration' has been provisionally accepted for publication in PLOS Computational Biology.

Best regards,

Roland R Regoes

Associate Editor

PLOS Computational Biology

Rob De Boer

Deputy Editor

PLOS Computational Biology

Reviewer's Responses to Questions

**Comments to the Authors:**

Reviewer #1: The authors have made a partial effort to comply with requests to shorten the manuscript. The manuscript has benefited greatly, but it is still excessively long. The length still partially obscures the justification for replacing the ID50 with the midSIN. The manuscript also retains the Spearman-Karber and Reed-Muench methods as a standard of comparison for the midSIN, and the bias in these methods is already known.

I remain supportive of the development of web calculators to replace the Spearman-Karber and Reed-Muench methods, but the manuscript's presentation of the midSIN is unlikely to promote the replacement much, if at all.

Reviewer #2: The paper is much improved, and I support is publication.

Reviewer #3: The authors have addressed my comments and, together with the other revisions, well improved the readability of the manuscript. I realised that my comments were not always as precise as I wished, but the authors well grasped their point and responded adequately. midSin is an easy to use tool that I will see to include in my groups activities.

**Have the authors made all data and (if applicable) computational code underlying the findings in their manuscript fully available?**

Reviewer #1: Yes

Reviewer #2: Yes

Reviewer #3: Yes

PLOS authors have the option to publish the peer review history of their article (what does this mean?). If published, this will include your full peer review and any attached files.

Reviewer #1: No

Reviewer #2: No

Reviewer #3: No

---

## [Editor Report · Acceptance letter]

13 Oct 2021

PCOMPBIOL-D-21-00213R1 

Time to revisit the endpoint dilution assay and to replace TCID₅₀ as a measure of a virus sample's infection concentration

Dear Dr Beauchemin,

I am pleased to inform you that your manuscript has been formally accepted for publication in PLOS Computational Biology. Your manuscript is now with our production department and you will be notified of the publication date in due course.

With kind regards,

Andrea Szabo
